# An Unusual Case of Serologically Confirmed Post-Partum Lyme Disease Following an Asymptomatic *Borrelia burgdorferi* Infection Acquired during Pregnancy and Lacking Vertical Transmission in Utero

**DOI:** 10.3390/pathogens13030186

**Published:** 2024-02-20

**Authors:** Charles S. Pavia, Maria M. Plummer, Alena Varantsova

**Affiliations:** 1Department of Biomedical Sciences, New York Institute of Technology, College of Osteopathic Medicine, Old Westbury, NY 11568, USA; 2Division of Infectious Diseases, New York Medical College, Valhalla, NY 10595, USA; 3Department of Clinical Specialties, New York Institute of Technology, College of Osteopathic Medicine, Old Westbury, NY 11568, USA; 4Department of Internal Medicine, College of Medicine, University of Central Florida, Orlando, FL 32308, USA

**Keywords:** Lyme disease, arthritis, pregnancy, *Borrelia burgdorferi*, Western blot, congenital transmission

## Abstract

In this report, we describe a 23-year-old female who, while pregnant, was exposed to *Borrelia burgdorferi* but did not develop significant signs or symptoms (joint pain, arthritis) of Lyme disease until shortly after delivering a healthy child at term. Serologic testing confirmed infection with *B. burgdorferi*. A 3-week course of treatment with doxycycline was completely curative. There was no evidence for congenital or perinatal transmission of this pathogen at any point pre-term or postnatally. The key reasons that could account for this unique clinical scenario are discussed in the context of previously published related reports.

## 1. Introduction

There have been relatively few well-documented cases of Lyme disease (also known as Lyme borreliosis), reviewed in [1,2,3,4] caused by several different sensu strictu strains of the spirochetal bacterium *Borrelia burgdorferi* occurring during pregnancy with potential congenital transmission. As a result, the transmission rate to the fetus and potential harmful effects were largely unknown for a long time [1,4]. More up-to-date information on this topic has become available in recently published detailed reports [2,3,4] that discuss the diagnosis, clinical course, and treatment of this spirochetal disease, with an emphasis on the pregnant patient. This information is especially important in light of our current understanding that, at least in the United States, the reported incidence of Lyme disease has been estimated to be greater than 470,000 cases per year [5], but this could be an overestimate if confirmatory Western blots were not included in the analysis. Previously published individual case reports (reviewed in ref. [6]) have suggested a possible association and adverse pregnancy outcomes; however, no specific pattern of teratogenicity has been shown with certainty, and a causal relationship has never been fully proven. In addition, a large epidemiological and serologic study [6] consistently failed to show pregnant women at an increased risk for developing more serious forms of extracutaneous Lyme disease, especially if appropriate antibiotic treatment is delayed. 

Unlike *Borrelia burgdorferi*, and to our knowledge, it has been known for a long time that there are other similarly related spirochetal pathogens that can be acquired congenitally, leading to severe adverse outcomes of the developing fetus, such as *Treponema pallidum*, the agent of syphilis [7], *Borrelia duttonii* and *B. recurrentis*, the causes of relapsing fever [8], which can cause spontaneous abortions, premature birth, and neonatal death. The maternal–fetal transmission of Borrelia is believed to occur either transplacentally or while traversing the birth canal and *Leptospira interogens*, the leptospirosis pathogen [9]. As far as we know, the only other disease that can be transmitted across the placental barrier by the same species of ticks (Ixodes) which transmit Lyme disease is babesiosis. Those data are quite solid, although congenitally acquired babesiosis is very rare [3].

In this report, we describe a favorable end-result birthing event, with no adverse outcomes of the fetus while in utero or to the newborn after birth, whose mother was a 23-year-old woman who became infected with *B. burgdorferi* and was only briefly symptomatic during her pregnancy. Somewhat surprisingly, the new mother went on to suddenly develop symptoms consistent with Lyme disease shortly after she delivered this healthy newborn. This account of the clinical case will be followed with a discussion on how our findings are related to some of the previously published reports involving women who presumably become infected with *B. burgdorferi* just prior to, or during, their pregnancy with or without having a serious impact on the developing fetus.

## 2. Case Report

On 30 January 2009, a 23-year-old primagravida female with no significant past medical history presented to a primary care office with the complaint of bilateral knee swelling and pain. She had delivered a healthy female three days previously. The patient stated that the pain and swelling gradually became worse over the next couple of days. Pain was reported as 8/10 in severity and was exacerbated by walking. She denied any falls, recent trauma or any birth-related injuries, or systemic symptoms such as fever or chills. She admitted to having a similar self-resolved episode of sharp bilateral knee pain without swelling occurring at 26 weeks (about 6 months) of gestation which lasted two days. She never sought medical attention for that very short episode. 

The pregnancy history was significant for an uncomplicated delivery at 41-week gestation with induced labor. There were no known prenatal complications. Prenatal travel history, however, was significant for several spring and summer trips to wooded areas in Narrowsburg, NY, USA. She denied any known exposure to ticks or the development of any rash. Upon further questioning of the patient, she did not recall that she or any of the other family members were bitten by a tick or had developed a skin rash during or shortly after these trips to Narrowsburg. She lived with her husband in Brooklyn, NY, USA, and was working and attending school at a local college while pregnant.

At presentation to the physician’s office, a CBC with differential, a complete metabolic panel and urinalysis, and an MRI were ordered, along with an ELISA for a *Borrelia burgdorferi screen*. Most of the non-serologic test results were normal except for a slightly below-average range for red blood cells (3.63; normal range = 3.92–5.13), hemoglobin (11.6; normal range = 12–15), and hematocrit (34.3; normal range = 36–48). An MRI revealed findings consistent with inflammation of the knee joints. The ELISA was positive, and a subsequently ordered Western blot showed multiple reactive bands (a total of eleven: Appendix A: supplemental) when probed with antibodies directed against the IgM isotype and the IgG isotype (18, 23, 28, 33, 41,43, 58, 66, and 93 kDa). These results met the CDC criteria for an infection with *B. burgdorferi.* All laboratory tests were performed by Quest Diagnostics (Teterboro, NJ, USA). The ELISA and Western blot kits used by Quest were manufactured by the Gold Standard (Freiburg, Germany}. 

The patient started a 3-week course of doxycycline, 300 mg orally, and she was sent to a rheumatologist for bilateral knee joint aspiration. Samples of the aspirates were submitted for microscopic analysis and revealed the presence of numerous white blood cells per field. Our patient reported significant improvement in the symptoms upon completion of the knee aspiration and the antibiotic course. She was able to return to baseline activities prior to the start of the symptoms. Long-term follow-up of the mother and the newborn showed no signs of mental or physical abnormalities or symptoms associated with Lyme disease. At periodic follow-up visits, the mother still had a positive *B. burgdorferi* ELISA, while the newborn maintained negative serology for *B. burgdorferi* antibodies. 

## 3. Discussion

### 3.1. Factors That Could Explain the Favorable Outcome of the Mother and Her Child

Narrowsburg is a hamlet in Sullivan County, New York, about 70 miles northwest of New York City. It lies along the Delaware river and is directly across from a Lyme disease-endemic region of the northeastern section of the state of Pennsylvania. In addition, according to data collected by the Vector-borne Disease Unit Bureau of Communicable Disease Control of the New York State Department of Health [10] and the Sullivan County Health Department [11] over the past several years, it has had a relatively low but consistent prevalence of ticks, though adjacent counties of Orange, Ulster and Delaware have a much higher prevalence [10]. Accordingly, there were certainly ample opportunities on numerous occasions for our patient to be exposed to *B. burgdorferi*-infected ticks during the spring and summer trips that she took there during her pregnancy. Since she resided in Brooklyn—one of the five boroughs of New York City—a non-endemic focus for tick-borne diseases, and she did not travel elsewhere, we feel confident that she acquired her infection during one of these trips possibly near or prior to mid-gestation. Even though our patient did not recall being bitten by a tick or developing an erythema migrans (EM) rash, these events often go unnoticed in a significant number (~20%) [12,13] of people who then progress to develop extracutaneous Lyme disease. Most interestingly, except for a very brief symptomatic period at 26 weeks gestation, she remained symptom-free (for Lyme disease or other tick-borne illnesses) for the remainder of the pregnancy. However, soon after delivering a healthy full-term female, she developed the classic signs of late-stage disease which include painful joint pain leading to a form of Lyme arthritis. A highly reactive Western blot performed by a certified testing facility showed that she developed a robust anti-*B. burgdorferi* antibody response with a pregnancy-associated delayed form of Lyme disease. Initially, when microscopic analysis of the synovial fluid was carried out by the diagnosing physician at that time, and no microorganisms were seen, it was still unclear to us why techniques in search of bacteria, such as *B. burgdorferi* or another potential bacterial pathogen, employing culture or a molecular method, were not performed. She had a rapid curative response to a standard antibiotic treatment regimen with doxycycline, which was one week less than the treatment regimen recommended by the Infectious Diseases Society of America for treating Lyme arthritis [13]. Given the time frame when she experienced her initial brief extracutaneous symptoms of joint pain 25 weeks (about 5 and a half months) before delivery, which resolved relatively quickly without medical attention, we estimate that her borrelial infection may have begun at least a few weeks earlier. We cannot also rule out the possibility that she may have unknowingly been infected on more than one occasion during the several trips to the area described or possibly with *B. miyamotoi*, although the serologic results tend to rule out this possibility. Additionally, while she became symptomatic soon after parturition, her newborn was symptom-free at birth and has never shown any of the usual signs or symptoms of active disease well into early childhood and beyond (Pavia, personal communication with the child’s mother, 2023). 

What could account for this unique clinical scenario which, to our knowledge, has rarely, if ever, been reported before? There are several realistic, plausible reasons that may explain this for either the mother and/or child. These are summarized in Table 1 and in more detail as follows:(1)Since many of the Lyme disease-associated symptoms [13,14] can be attributed to a Borrelia-elicited immune/inflammatory-mediated reaction rather than a direct attack on host cells by a toxin or other virulence factor, one possible mechanism could include maternal downregulation of certain immunologic responses during various stages of gestation [15,16,17,18,19]. Supporting this possibility are the well-documented findings, some of which are based on the pioneering work of Stites, Siiteri, and Pavia, outlined more than 40 years ago [16,17,18], showing that certain pregnancy-associated hormones, especially those produced by placental trophoblastic cells, may exert various immune-modulating properties [16,17]. These include human chorionic gonadotropin (hCG) [20,21], human placental lactogen (hPL) [21], estrogens, progestogens (especially progesterone) [22], relaxin [23,24], and estradiol [16,17,25]. As soon as the pregnancy ends, these hormones no longer operate at immune-inhibitory/regulatory concentrations, thus allowing for the re-emergence of a previously inactive infection or a dormant inflammatory state or other immunomodulatory activities [26,27]. Years later, these initial and breakthrough findings were subsequently extended in studies using animal infection models of Lyme disease, where it was shown that gestational attenuation of Lyme arthritis is mediated by progesterone and interleukin-4 [28], and that IL-10 can play a dual role in murine Lyme disease by regulating the severity of arthritis and host defense [29].(2)A possible protective mechanism for the newborn pertains to the potential phagocytic/borreliacidal activity and immunocompetence of the cells located on the surface and internally of the placental trophoblast within the placenta as it matures and grows, as previously reported [30,31]. It was shown that, when cultured in vitro, mouse trophoblast cells were able to phagocytize and kill malaria parasites; in addition, placental lymphocytes from Balb/c mice expressed a weak-to-moderate proliferative response to concanavalin A (Con A), phytohemagglutinin (PHA), and pokeweed mitogen when compared to the mitogenic responses of adult spleen cells. The stimulatory effects of Con A and PHA were abrogated after depleting the T cells from the placental lymphocyte preparation. Lipopolysaccharide-induced reactivity was similar for both cell populations.

### 3.2. Evidence for Either in Utero Transmission or Lack of B. burgdorferi during Pregnancy

Currently, there is insufficient evidence, based on epidemiological research reviewed in [1,2,3,4], to rule out the possibility that uncommon abnormal manifestations of Lyme disease can occur during pregnancy as compared to their occurrence in the general non-pregnant population. This has yet to be reported by the obstetrical coam amunity. Our patient and her newborn child did not manifest these. Both mother and child remained healthy long after parturition and successful treatment of the mother with doxycycline, and subsequently, during routine follow-up physician visits. And now, many years later, both of them have remained healthy, showing no signs or symptoms commonly associated with Lyme disease (Pavia, personal communication with the mother, 2023). 

Our findings are in marked contrast to some of the earlier related work in this area [32,33,34,35]. In one of these studies [33], processed fetal autopsy specimens, obtained from stillborn fetuses of four pregnant women at various gestational stages (whose Lyme disease status was unknown), were analyzed using the Warthin–Starry silver stain technique after attempting to culture various tissue samples in modified Kelly’s medium. Few spirochete-like forms could be seen using darkfield microscopy, unlike what the CDC has shown for an infected adult cardiac tissue sample [36]. This suggests that it is biologically plausible for a spirochete, such as *B. burgdorferi*, to be visible microscopically with standard histopathology procedures after being vertically transmitted to the fetus and colonizing fetal heart tissue. Results were less impressive, however, when an indirect immunofluorescent staining technique and polyclonal antibodies derived from the presumed infected mother (supposedly directed against *B. burgdorferi*) were used as the probes. Also, under these conditions, presumed spirochetal forms became partially obscured and could not be readily seen. This also occurred when a matched “control culture” of the well-established B31 strain was prepared simultaneously. Even many of these in vitro-derived “control” structures lacked the typical helical or corkscrew shape of spirochetes and some of them could have been remnants of dead, degenerating or inactive organisms. It would have been better if these investigators had also used an already well-characterized and more reliable serum source having high-titer anti-borrelia antibodies, typically acquired from an appropriately immunized laboratory animals (often rabbits), to test for reactivity to the already known and presumed isolated pathogen, thereby optimizing the possibility of identifying authentic Lyme disease spirochetes in these situations. If not available commercially, these types of antibodies can usually be produced under relatively routine experimental laboratory conditions. For many years, they have been used for numerous purposes before the emergence of even more specific monoclonal antibodies, for the purpose of optimizing the possibility of identifying Lyme disease bacteria, as well as in other unrelated studies. In addition, and unfortunately, another confounding variable was encountered whereby many of these cultures contained various bacterial contaminants of unknown species. 

It is important to point out that the development of more recent and reliable advancements in diagnostic testing procedures over the past 25–30 years with highly specific detection levels, such as readily available monoclonal antibodies of high specificity, gene probes and PCR, as well as improvements in the two-tier serologic algorithm involving Western blots, would have provided more convincing evidence for a possible vertical transmission. Unfortunately, these types of reagents were unavailable at the time that these studies were taking place. Also, some of the aforementioned studies have been unable to consistently define any characteristic pathological effect of *B. burgdorferi* infection in the fetus relative to what typically occurs outside of pregnancy which has been well established with other pathogens, such as *Treponema pallidum*, *Listeria monocytogenes*, *Borrelia duttonii* [8], and *B. recurrentis*, *Neisseria gonorrhoeae*, *Toxoplasma gondii*, and parvovirus B19 [7]. These organisms are typically known to be able to pass through the placenta and reach the fetus at various gestational stages or perinatally as the baby passes through the birth canal. These may cause serious tissue and organ damage, sometimes leading to fetal wastage or death. With regard to congenital toxoplasmosis, fetal death or major abnormalities such as blindness and cognitive impairment may occur when infection is acquired during the first trimester [7]. This raises the possibility that if congenital transmission of *B. burgdorferi* capable of causing serious tissue/organ occurs, it may be gestational-stage specific. A somewhat-related case study [37] involved a woman who developed “documented” Lyme disease during the first trimester of pregnancy but did not receive antibiotic therapy. She delivered the baby at 35 weeks gestation who died of congenital heart disease during the first week post-partum. With the exception of the heart, histologic examination of autopsy material from various other tissue/organ sites showed a few irregular-looking organisms that were interpreted as being “morphologically compatible with the Lyme disease spirochete”. Many oil immersion fields were examined in trying to identify the spirochete; more than one was never considered as being *B. burgdorferi*. As with many other related reports, these structures, however, did not resemble the typical and unique shape of *B. burgdorferi* (Figure 1), which one of our research groups [38,39,40] as well as other investigators [41] have consistently observed and documented after culturing blood or biopsied skin material from suspected non-pregnant Lyme disease patients.

In addition, another report [2] has summarized some of the limited proof for a possible transplacental transmission of *B. burgdorferi* from a few other case studies, but inconsistent evidence was also provided for adverse birth outcomes associated with pregnancy-associated presumed Lyme disease. This systematic review on the impact of gestational Lyme disease in humans on the fetus and newborn revealed that mainly uncommon adverse outcomes for the fetus were occurring and may depend on the stage of the pregnancy. It is noteworthy that this report also described a clinical case that closely resembled ours in that there was a favorable outcome in a 42-year-old woman who developed Lyme disease in the third trimester and was treated with a full course of oral amoxicillin.

Most recently, a written questionnaire survey-type report was published [42], attempting to lay the foundation for trying to make any correlation between Lyme disease and pregnancy outcomes. Nearly 1000 participants from a large international cohort of pregnant and non-pregnant women responded to the questionnaire and supposedly lived mostly in certain Lyme disease-endemic areas of North America (northeastern coastal areas of the USA stretching south from Maine to northwestern Virginia and three distinct localized areas of eastern, central and western Canada) [5,43] and certain undefined areas of Europe. However, the exact locations were not revealed. Here is where the study investigators categorized respondents unconventionally with having either “probable treated”, “probable untreated”, “possible untreated”, “no evidence of”, Lyme disease, or “unclear”. However, it was not fully obvious how these determinations were made, and they may have been influenced by a phenomenon known as “recall bias”. 

Although a considerable amount of data were obtained, several factors make their overall interpretation problematic or too imprecise. This limits how the results can be applied to future studies in trying to better understand what may be highly complex interactions between borrelial-infected pregnant women and their developing feto-placental unit. Perhaps two of the most alarming and serious deficiencies were that: (i) a significant number (nearly 40 of the participants) indicated that they self-diagnosed an EM rash but there was no indication that this was confirmed by a knowledgeable healthcare provider. Even still, what looks like a bona fide EM rash could easily be mistaken for the closely resembling erythema marginatum-type rashes [44] that have been linked to mostly adverse drug reactions and infections with certain pathogens, such as the herpes virus and the bacterium *Mycoplasma pneumoniae*. A case in point regarding this matter can be traced to a recently published well-controlled study [45] that included some retrospective analysis. This provided very meaningful data involving 604 pregnant Slovenian women (at various gestational stages) who received the prevailing course of antibiotic treatment soon after a physician confirmed the skin lesion as being EM. The results showed that 86% of them had favorable outcomes whereas the remaining 14% did not. However, some of those from this latter group could not be traced to an inadequate treatment regimen but were rather associated with other non-Lyme-related gestational anomalies, and for some of the respondents, the required serologic confirmation based on Western blot testing was performed, but it was unclear whether they were performed in a certified diagnostic environment using only FDA-approved, or equivalent, typically commercially available test kits that included Western blots. It is also very important that the CDC criteria were followed for interpreting the banding patterns. In this regard, more recently, this issue has been rigorously studied [46,47,48,49,50] and showed that it can be problematic, mostly due to inadequate specificity, or by misinterpretation on the meaning of some of the banding patterns by inexperienced personnel performing the blots.

In summary, the evidence gathered over the past 35 years regarding whether or not a borrelial infection contracted during pregnancy could have serious consequences to the mother and/or the fetus has been inconsistent and, at times, controversial [4,33,35], until recently, where Lyme disease is now reportable in many jurisdictions (to state health departments, the CDC) and with much better, more sensitive detection methods, and we now have the opportunity to further explore this topic in terms of providing clinicians more clear-cut evidence-based data on how to best manage a woman who may have contracted Lyme disease while pregnant in such a way that ensures a favorable outcome. 

Nonetheless, despite suggestions by others [42] and in partial agreement with us, the global evidence that has been accrued on pregnancy-associated Lyme disease, ever since Lyme disease became reportable to public health authorities, does not fully characterize the potential impact of gestational Lyme disease. Future research which addresses the knowledge gaps in this area may change the overall impact and meaning of these findings that have been accrued over the past 35 years. Thus, there are still significant knowledge gaps about the relationship of *B. burgdorferi* infection and possible adverse birth outcomes [1,2,3,4,6,42]. This is especially true in terms of whether any possible negative outcome of the fetus is dependent on which trimester/gestational stage that the prospective mother is infected with the Lyme disease bacterium, and whether it is strain-dependent for this pathogen since different geographical isolates of *B. burgdorferi* can be categorized uniquely based on using sophisticated molecular techniques [38,43] such as restriction fragment length polymorphism associated with their 23S–25S gene cluster [50]. 

Another important consideration is whether there are any co-infections with other tick-borne pathogens, such as Babesia, Anaplasma, Ehrlichia organisms, Powasson virus (only North America) or tick-borne encephalitis virus (Europe only). Along these lines, our patient showed the first sign/symptom of a possible Lyme disease-related abnormality (knee joint pain of a 2-day duration) at the 26th week of her pregnancy that went unattended. Pinpointing when she was likely bitten by an infected tick, perhaps more than once, is not possible since our patient took multiple trips during the spring and summer seasons to a relatively low-endemic area which was the only possible site where she would have been exposed to infected ticks. However, some estimate could be made if we extrapolate from the time course for the initial appearance of extracutaneous signs/symptoms of Lyme disease that typically appear in the general, non-pregnant population after a tick bite. Combining our patient’s account of her travel history during her pregnancy, it is possible that she was infected at least once sometime prior to the 22nd week of her pregnancy but possibly even much earlier. This calculation is based on the time frame that has been established from when an extracutaneous abnormality, such as arthritis, first appears after a tick bite or when an EM rash is recognized [12,13,14]. We cannot also overlook the possibility that she may have been infected/coinfected with the spirochete, *B. miyamotoi* (although unlikely) which shares some of the same epidemiologic profile as *B. burgdorferi* [43] having been found in ticks in 19 states, with the infection prevalence being 0.5–3.2%. Ticks tested in 20 additional states were all negative. *B. miyamotoi* has been detected in all Canadian provinces, excluding Newfoundland, in *I. scapularis ticks*. The most common symptoms are fever, chills, and headache. Other common symptoms include body and joint pain and fatigue. Rashes are uncommon, with fewer than 1 in 10 patients developing a rash [51].

Given these uncertainties pertaining to the impact that *B. burgdorferi* might have on the developing fetus and the consistent evidence showing that fewer adverse birth-related abnormalities occur if Lyme disease is promptly treated, physicians should continue to remain thorough in their diagnosis and treatment of Lyme disease in pregnant women. In addition, more research is warranted toward defining more completely any knowledge gaps in this area of great concern. As suggested by others [42], and we partially agree, that the global evidence does not fully characterize the potential impact of gestational Lyme disease, and future research that addresses the knowledge gaps that still exist may change the overall impact and meaning of these findings, especially in terms of whether any possible negative outcomes to the fetus are dependent on which trimester/gestational stage that the prospective mother gets infected with *B. burgdorferi*.

Following this train of thought, should women who live in a Lyme disease-endemic area and are planning to become pregnant, or who are pregnant, be periodically monitored for an ongoing *B*. *burgdorferi* infection through periodic serologic testing similar to the testing for toxoplasmosis that has been taking place in France since 1978? [45]. Perhaps the obstetrics community and the CDC need to agree on making a decision along these lines on this matter. In conclusion, in view of the lack of sufficient and clear-cut case study reports showing significant adverse outcomes for either a woman who develops Lyme disease during her pregnancy or her fetus, confirmed by either verifiable culture and/or more sophisticated and up-to-date molecular or serologic testing techniques, it remains unclear as to how frequent this clinical scenario occurs or has occurred in the past. Perhaps with a few exceptions, studies showing a definitive link between gestational Lyme disease and an increased risk of maternal and/or congenital anomalies are lacking. Consistent with this lack of fully verifiable criteria, there are other concerns as pointed out by others [3]: “Indeed, the wide variety of anomalies described and the lack of a specific pattern of teratogenicity argue against a causal link.” In this report, we provide substantial evidence of a pregnant woman who was likely exposed to the Lyme disease spirochete at an undetermined gestational stage and did not receive any antibiotic treatment during her pregnancy, yet she only became significantly symptomatic after giving birth to a heathy child who showed no signs of having congenitally acquired a *B. burgdorferi* infection. Is this an isolated occurrence? Perhaps it is, especially since, after an intense PubMed search of the published literature on this topic, we were unable to find a comparable clinical case study resembling ours. At least the good news coming from our clinical case report is that, with many years having now passed since the child was born, both mother and child continue not to show any signs or symptoms of Lyme disease or any other tick-borne disorder (Pavia, 2023, personal communication with the mother). We have also discussed some of the likely microbiologic/immunologic parameters that can explain these phenomena. In addition, given the uncertainties as to whether women living in an endemic area who are pregnant or plan to be pregnant, and could also potentially be infected with the Lyme bacterium, consideration should be given for periodic testing at designated gestation stages using approved detection systems, such as serology, for detecting anti-*B. burgdorferi* antibodies.

## Figures and Tables

**Figure 1 pathogens-13-00186-f001:**
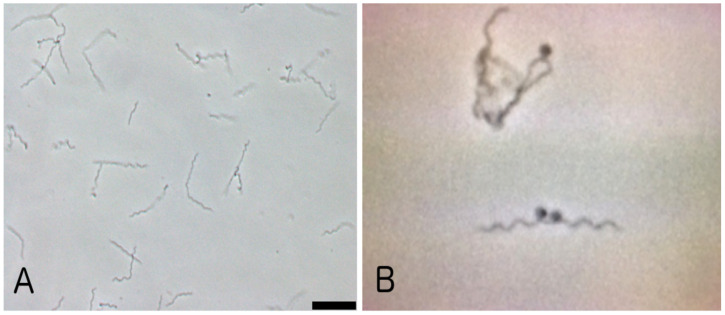
Photomicrographs of a maintenance culture of the strain BL206 of *B. burgdorferi* derived from a blood culture of a Lyme disease patient seen at the New York Medical College Lyme Disease Clinic in 1999 [38]; (**panel A**), phase contrast microscopy, at 200× magnification; (**panel B**), phase contrast microscopy at 500× magnification. It is noteworthy that these types of images, representative of *B. burgdorferi*’s unique shape, have rarely, if ever, been reported in the various past case study reports claiming fetal damage due to a congenitally acquired infection.

**Table 1 pathogens-13-00186-t001:** Factors that could explain why our *B. burgdorferi*-infected patient was asymptomatic during most of her pregnancy and then developed a form of Lyme arthritis, but not her newborn child, ^a^ soon after parturition.

Factor	Relevant References
Immunologic regulation caused by pregnancy-associated hormones ^a^	[16,17,18,19,20,21,22,23,24,25,29]
Phagocytic/microbicidal activity by placental trophoblastic cells ^b^	[31]
Immunocompetence of placental lymphocytes ^b^	[30]

^a^ Pertains to the mother only. ^b^ Pertains to the non-vertical transmission of Borrelia to the newborn.

## Data Availability

The raw data supporting the conclusions of this article will be made available by the authors on request.

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
