# Peer review of "An Unusual Case of Serologically Confirmed Post-Partum Lyme Disease Following an Asymptomatic Borrelia burgdorferi Infection Acquired during Pregnancy and Lacking Vertical Transmission in Utero"

_pathogens, 2024, doi:10.3390/pathogens13030186_

Round 1

Reviewer 1 Report

Comments and Suggestions for Authors

It is no surprise at all that Lyme disease in pregnancy did not result in an adverse birth outcome. This is reviewed elsewhere in the literature as well as in the IDSA-AAN-ACR guidelines. Even to whatever microscopic degree Borrelia has been found in placental specimens, there has never been an accepted Lyme-associated congenital syndrome and the standard of care does not dictate any different management of a pregnant vs non-pregnant woman with Lyme disease (except with respect to the choice of doxycycline). It is to be expected that this patient would have a normal birth outcome. Given the incidence of Lyme disease (though probably NOT > 400,000 as the authors cite, as that figure is based only on claims data and not verified clinical data), Lyme disease and exposure in pregnancy is probably quite common in endemic areas.

So while I understand that it's interesting that a woman would present immediately after pregnancy, the birth outcome is not particularly interesting. What would be more interesting would be to reframe the article about why she didn't manifest until after delivery. Was it the relative immunosuppression of pregnancy? Did fluid retention mask knee arthritis? I think that is a more interesting discussion than the birth outcome.

Reviewer 2 Report

Comments and Suggestions for Authors

Overall, this is an important case report with a review of the pertinent literature and critique of other papers on this subject. But there are several minor weaknesses, and the content of the discussion needs substantial copy-editing and reduction in length.

(1) Introduction, line 26: Lyme disease is caused by different species of B. burgdorferi sensu lato.  If the authors mean to use "strains", then restriction to one species, such as B. burgdorferi sensu stricto, would be more appropriate.

(2) Introduction, lines 33-34: The use of the word "reported" could be confusing. The number of "reported" cases, i.e. recorded as cases by state health departments and ultimately CDC, is much less than 470,000 per year.  That is an estimate based on extrapolations.

(3) Introduction, line 40: "increased risk" in this context should be clarified.  Do the authors mean increased risk of acquiring the infection, of having a more severe course if infected, or of effects on the fetus or pregnancy outcome compared to women who are not pregnant.

(4) Introduction: lines 42-46: The sentence is awkward and needs restructuring.  In addition, there are other species that cause relapsing fever (no need to add "borreliosis") that affect the pregnant woman and are congenitally transmitted. These also include Borrelia recurrentis, the cause of louse-borne relapsing fever, and Borrelia crocidurae, another cause of soft tick relapsing fever. For louse-borne relapsing fever one recent comprehensive review was by P. Kahig et al. in PLoS Neglected Tropical Diseases in 2021. lThe authors should also include some comment about Borrelia miyamotoi, which is a cause of hard tick relapsing fever, and is transmitted by same ticks as those that transmit Lyme disease, etc. 

(4) Lines 75-84: The lab results should be clarified in the text here.  This is critical information. What are the units for the CBC values?  It probably should be "Borrelia burgdorferi" ELISA and not "Lyme" ELISA.  Was it whole cell or C6 peptide or some combination? Was it IgG or IgM or both for ELISA and Western blot? 

(5) Line 86: What was the dose of doxycycline? Presumably it was oral.

(6) Line 89: The authors should clarify what is meant by "no bacteria were identified". This was probably a standard laboratory culture for a joint aspirate and not culture or PCR for B. burgdorferi.  

(6) Lines 93-94: If possible the authors should present the results of the serologic assays of the woman and the child with dates in a table.  

(7) The authors should acknowledge that the patient might also have had Borrelia miyamotoi infection as well as B. burgdorferi infection, and discuss using the serology results why that was unlikely.  In any case, some brief discussion of the risk--if any known--of congenital or perinatal infection from B. miyamotoi is warranted.  

(8) The discussion is pretty long and could benefit from some copy-editing and a focus on the essential elements of the report.  See comment below.

(9) Lines 157-160: The authors could be more clear what they mean here. Readers should not have to look up the references to understand what the mechanism might be.

(10) Lines 161-177: This section is very hard to follow. There is also something missing at the end. Not clear what "strain-dependent" means here. Do the authors intend to distinguish different species of bacteria rather than strains within a species.  It is "adhesins", not "adhesions".  And what would be specific about Borrelia burgdorferi here that would differentiate from B. duttonii/B. recurrentis which do cross the barrier and infect the fetus?  This could be elaborated on beyond "not operative".  

(11) Table 1. B. burgdorferi is not wider than relapsing fever Borrelia species and can be just as short.  So why is B. burgdorferi singled out here?  This was not very convincing.  I suggest dropping this line in the table.  And then is the table needed at all?

(12) The extended discussion and critique of reports claiming worse outcomes of pregnancies with Lyme disease is a very useful contribution to the literature, but the same arguments could be made more economically. The content could be reduced by half without affecting the impact. Substantial copy editing is recommended. 

Comments on the Quality of English Language

See comments above about the length of the discussion and the need for copy editing. 
